# Shape Arithmetic Expressions

**Krzysztof Kacprzyk**
University of Cambridge
kk751@cam.ac.uk

**Mihaela van der Schaar**
University of Cambridge
mv472@cam.ac.uk

## Abstract

Symbolic regression has excelled in uncovering equations from physics, chemistry, biology, and related disciplines. However, its effectiveness becomes less certain when applied to experimental data lacking inherent closed-form expressions. Empirically derived relationships, such as entire stress-strain curves, may defy concise closed-form representation, compelling us to explore more adaptive modeling approaches that balance flexibility with interpretability. In our pursuit, we turn to Generalized Additive Models (GAMs), a widely used class of models known for their versatility across various domains. Although GAMs can capture non-linear relationships between variables and targets, they cannot capture intricate feature interactions. In this work, we investigate both of these challenges and propose a novel class of models, Shape Arithmetic Expressions (SHAREs), that fuses GAM's flexible shape functions with the complex feature interactions found in mathematical expressions. SHAREs also provide a unifying framework for both of these approaches. We also design a set of rules for constructing SHAREs that guarantee transparency of the found expressions beyond the standard constraints based on the model's size.

## 1 Introduction

**Symbolic Regression.** Symbolic regression (SR) is an area of machine learning that aims to construct a model in the form of a *closed-form expression*. Such an expression is a combination of variables, arithmetic operations $(+, -, \times, \div)$, some well-known functions (trigonometric functions, exponential, etc), and numeric constants. For instance, $3\sin(x_1 + x_2) \times e^{2x_3^2}$. Such equations, if concise, are interpretable and well-suited to mathematical analysis. These properties have led to applications of SR in many areas such as physics [37], medicine [4], material science [48], and biology [12]. Symbolic regression is usually validated on synthetic datasets with closed-form ground truth equations [44, 32, 6]. However, as we investigate in Section 2.1, closed-form functions are often inefficient in describing some relatively simple relationships producing overly long expressions. They are also not compatible with categorical variables. Further discussion on symbolic regression methods can be found in Appendix D.

**Generalized Additive Models.** Widely used transparent models are Generalized Additive Models (GAMs) [17, 25]. They model the relationship between the features $x_i$ and the label $y$ as

$$g(y) = f_1(x_1) + \ldots + f_n(x_n) \qquad (1)$$

where $g$ is called the *link functions* and the $f_k$'s are called *shape functions*. These models allow arbitrary complex shape functions but exclude more complicated interactions between the variables. They are deemed transparent as each function $f_k$ can be plotted, and thus the contribution of $x_k$ can be understood. Extensions of GAMs that include pairwise interactions have also been proposed (GA$^2$M) [26]. In these settings, the model can contain 2D shape functions that can be visualized using a heatmap. The main disadvantage of such models is their inability to model more complicated interactions, for instance, $\frac{x_1 x_2}{x_3}$ (see Section 2). See Appendix D for more discussion about GAMs.

NeurIPS 2023 AI for Science Workshop.

**Transparency of closed-form expressions.** A model is considered *transparent* if by itself it is understandable—a human understands its function [5]. The transparency of symbolic regression can be compromised if the found expressions become too complex to comprehend. In many scenarios, an arbitrary closed-form expression is unlikely to be considered transparent. Most of the current works limit the complexity of the expression by introducing a constraint based on, e.g., the number of terms [41], the depth of the expression tree [13], or the description length [43]. Although these metrics often correlate with the difficulty of understanding a particular equation, size does not always reflect the equation's complexity as it does not focus on its semantics. Some recent works introduce a recursive definition of complexity that takes into account the type and the order of operations performed [46, 23]. Although they are a step in the right direction, they are not grounded in how the model will be analyzed, and thus, it is not clear if they capture how comprehensible the model is (further discussion in Appendix D).

**Contributions and outline.** In Section 2, we investigate the limitations of SR and GAMs. In Section 3, we introduce a novel class of models called **SH**ape **AR**ithmetic **E**xpressions (SHAREs) that combine GAM's flexible shape functions with the complex feature interactions found in closed-form expressions thus providing a unifying framework for both approaches. In Section 4, we introduce a new kind of transparency that goes beyond the standard constraints based on the model's size and apply it to SHAREs. We also investigate theoretical properties of transparent SHAREs. Finally, we demonstrate their effectiveness through experiments in Section 5.

## 2 Limitations of current approaches

### 2.1 Symbolic regression struggles with non-closed-form expressions.

Symbolic regression excels in settings where the ground truth is a closed-form expression [44]. However, its effectiveness becomes less certain when applied to scenarios with no underlying closed-form expressions. Some phenomena do not have a closed-form expression (e.g., non-linear pendulum), and many functions in physics are determined experimentally rather than derived from a theory and are not inherently closed-form (e.g., current-voltage curves, drag coefficient as a function of Reynolds number, phase transition curves). This is even more relevant in life sciences, where the complexity of the studied phenomena makes it more difficult to construct theoretical models. We claim symbolic regression struggles to find a compact expression for some relatively simple univariate functions.

**Example: stress-strain curves.** To illustrate our point, we try to fit a symbolic regression model to an experimentally obtained stress-strain curve. We use data of stress-strain curves in steady-state tension of aluminum 6061-T651 at different temperatures obtained by [1]. Figure 1 (left panel) shows a sample of these curves. These functions are relatively simple as they can be divided into a few interpretable segments representing different behaviors of the material. We use a symbolic regression library PySR [13] to fit the stress-strain curve of aluminum at 300°C. We fit the model and present some of the found expressions in Figure 1 (right panel). The size of a closed-form expression is defined as the number of terms in its representation. For instance, $\sin(x+1)$ has complexity 4 as it contains four terms: $\sin$, $+$, $x$, and 1. We can see that small programs do not fit the data well. A good fit is achieved only by bigger expressions. However, such expressions are much less comprehensible, and thus, their utility is diminished.

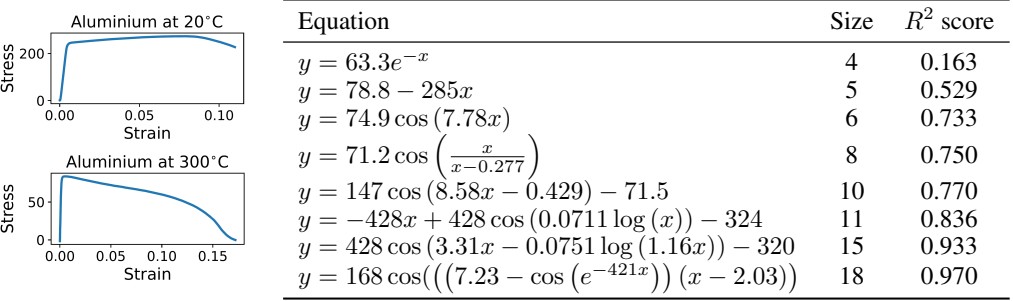

| Equation | Size | $R^2$ score |
| --- | --- | --- |
| $y = 63.3e^{-x}$ | 4 | 0.163 |
| $y = 78.8 - 285x$ | 5 | 0.529 |
| $y = 74.9\cos{(7.78x)}$ | 6 | 0.733 |
| $y = 71.2\cos\left(\frac{x}{x-0.277}\right)$ | 8 | 0.750 |
| $y = 147\cos{(8.58x - 0.429)} - 71.5$ | 10 | 0.770 |
| $y = -428x + 428\cos{(0.0711\log{(x)})} - 324$ | 11 | 0.836 |
| $y = 428\cos{(3.31x - 0.0751\log{(1.16x)})} - 320$ | 15 | 0.933 |
| $y = 168\cos(((7.23 - \cos{(e^{-421x})})(x - 2.03))$ | 18 | 0.970 |

Figure 1: Left panel: examples of stress-strain curves. Right panel: Some of the equations discovered by Symbolic regression when fitted to the stress-strain curve of aluminum at 300°C.

## 2.2 GAMs cannot model complex interactions

The main disadvantage of GAMs is that they are poor at modeling more complicated, non-additive interactions (involving 3 or more variables). Such interactions occur frequently in real life. For instance, many equations from physics involve multiplying a few variables together. To illustrate this point, we choose a few simple equations from the Feynman Symbolic Regression Database [44] and compare the performance of GAMs and GA$^2$Ms with a black-box machine learning model. We implement GAMs and GA$^2$Ms using Explainable Boosted Machines (EBMs) [25, 26] and choose XGBoost [11] for a black-box model. We choose equations so that they represent a variety of non-additive interactions between variables (see Table 1). For a detailed discussion, see Appendix D.

**Results.** We report the results in Table 1. The performance of GAMs is much lower than the performance of a full-capacity model (whose $R^2$ score is close to 1.0 as no noise was added to the dataset). The gap between GAM and XGBoost is partially closed by adding pairwise interactions in GA$^2$Ms. This dramatically improves the score in some cases (e.g., equation I.8.14) but still under-performs in others (e.g., equation I.32.5). It is important to note that pairwise interactions decrease the comprehensibility of the model. In particular, 2D heatmaps are more challenging to understand than plots of univariate functions, and the individual shape functions cannot be analyzed independently. As the shape functions have overlapping sets of arguments, we may have to analyze many shape functions at the same time to understand the model.

Table 1: Performance of additive models (GAM and GA$^2$M) compared to a full capacity model (XGBoost) on datasets from the Feynman Symbolic Regression Database containing complex variable interactions. We show the mean $R^2$ score and a standard deviation in the brackets.

| Eq. Num | Equation | GAM | GA$^2$M | XGBoost |
|---|---|---|---|---|
| I.6.20b | $f = e^{-\frac{(\theta-\theta_1)^2}{2\sigma^2}}/\sqrt{2\pi\sigma^2}$ | 0.731 (.010) | 0.896 (.004) | 0.997 (.000) |
| I.8.14 | $d = \sqrt{(x_2-x_1)^2 + (y_2-y_1)^2}$ | 0.229 (.011) | 0.966 (.000) | 0.989 (.000) |
| I.12.2 | $F = \frac{q_1 q_2}{4\pi\epsilon r^2}$ | 0.676 (.011) | 0.950 (.003) | 0.993 (.001) |
| I.12.11 | $F = q(E_f + Bv\sin(\theta))$ | 0.675 (.004) | 0.955 (.001) | 0.996 (.000) |
| I.18.12 | $\tau = rF\sin(\theta)$ | 0.760 (.002) | 0.981 (.000) | 0.999 (.000) |
| I.29.16 | $x = \sqrt{x_1^2 + x_2^2 - 2x_1 x_2 \cos(\theta_1 - \theta_2)}$ | 0.298 (.007) | 0.902 (.002) | 0.983 (.001) |
| I.32.5 | $P = \frac{q^2 a^2}{6\pi\epsilon c^3}$ | 0.444 (.015) | 0.835 (.009) | 0.988 (.001) |
| I.40.1 | $n = n_0 e^{-\frac{magx}{k_b T}}$ | 0.736 (.003) | 0.899 (.003) | 0.981 (.001) |
| II.2.42 | $P = \frac{\kappa(T_2 - T_1)A}{d}$ | 0.615 (.006) | 0.937 (.002) | 0.990 (.000) |

## 3 Shape Arithmetic Expressions

In this section, we introduce a new type of machine learning model that connects symbolic regression's ability to model interactions with GAM's power of efficiently describing univariate functions by plots. This new family of models addresses the issues of both GAMs and symbolic regression that we discussed in the previous section.

Inspired by the GAM literature, we define a set of *shape functions* $\mathcal{S}$, where each $s \in \mathcal{S}$ is a univariate function $s : \mathbb{R} \to \mathbb{R}$. This might be, for instance, a set of cubic splines or univariate neural networks. Let $\mathbb{B} = \{+, -, \div, \times\}$ be a set of binary operations. Let us denote real variables as $x_i$. We introduce *Shape Arithmetic Expression* (SHARE) as a mathematical expression that consists of a finite number of shape functions, binary operations, variables, and numeric constants. For instance, see Equation 2, where $s_1, s_2, s_3 \in \mathcal{S}$ are the shape functions and need to be plotted next to the equation to understand the whole model. Formally, we represent SHAREs as expression trees (types of graphs) where each node is either a binary operation $b \in \mathcal{B}$ (with two children), a univariate function $s \in \mathcal{S}$ (with one child), a variable or a numeric constant (as leaves). Equation 2 represented as a tree can be seen in Figure 2. We borrow the terminology from SR literature and define the complexity (or size) of a SHARE as the number of nodes in its expression tree. The depth of a SHARE is defined as the depth of its expression tree.

*Remark* 1. Any GAM is an example of a SHARE. If we choose $\mathcal{S}$ to be a set of some well-known functions (e.g., $\mathcal{S} = \{\sin, \cos, \exp, \log\}$) then closed-form expression can also be considered

SHAREs. In general, however, $\mathcal{S}$ is supposed to be a flexible family of functions that are fitted to the data and are meant to be understood visually.

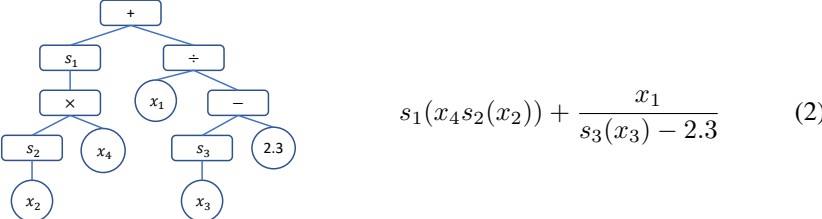

$$s_1(x_4 s_2(x_2)) + \frac{x_1}{s_3(x_3) - 2.3} \qquad (2)$$

Figure 2: Shape Arithmetic Expression represented as a tree.

**Why univariate functions?** We decided to only use univariate functions for two reasons: they are easy to understand, and they are sufficient. Firstly, they are easy to comprehend because they can always be plotted. While analyzing them, we have to keep track of only one variable, and we can characterize them using monotonicity. Univariate functions are also much easier to edit in case we want to fix the model. Secondly, the Kolmogorov–Arnold representation theorem [22] states that for any continuous function $f : [0, 1]^n \to \mathbb{R}$, there exist univariate continuous functions $g_q . \phi_{p,q}$ such that

$$f(x_1, \ldots, x_n) = \sum_{q=0}^{2n} g_q \left( \sum_{p=1}^{n} \phi_{p,q}(x_p) \right) \qquad (3)$$

That means in principle, for expressive enough shape functions, SHAREs should be able to approximate any continuous function. However, SHAREs of that form would not necessarily be very transparent. We discuss the transparency of SHAREs in the next section.

## 4 Transparency

As explained in Section 1, the transparency of symbolic regression can be compromised if the found expressions become too complex to comprehend. In many scenarios, an arbitrary closed-form expression is unlikely to be considered transparent. To see that, it is enough to realize that any fully connected deep neural network with sigmoid activation functions is technically a closed-form expression. As SHAREs extend SR, they inherit the same problem. Current works introduce constraints that are not grounded in how the model will be analyzed; thus, it is unclear whether they capture how comprehensible the model is. That includes constraints based on model size [41, 13, 43] and even recent semantic constraints [46, 23] (further discussion on SR constraints in Appendix D). We take an alternative approach. We define transparency implicitly by proposing two general rules for building machine learning models in a transparency-preserving way, and we justify why they may be sufficient for achieving transparency in certain scenarios. These rules, in turn, allow us to define a subset of transparent SHAREs.

**Rule 1.** Let $s$ be any univariate function. $s(x_i)$ is transparent, where $x_i$ is any variable. If $f$ is transparent then $s \circ f$ is also transparent.

**Rule 2.** Let $b \in \mathcal{B}$ be a binary operation. If $f$ and $g$ are transparent and have disjoint sets of arguments then $b \circ (f, g)$ is also transparent.

Motivated by research on human understanding and problem solving [29, 39, 28, 40], we assume that in some scenarios *understanding a complex expression involves decomposing it into smaller expressions and understanding them and the interactions between them.* Thus the model can be understood from the ground up. This is in agreement with recent research in XAI that highlights *decomposability* as a crucial factor for transparency, enabling more interpretable and explainable machine learning methods [5]. The rules we propose offer a rigorous way to encode this criterion for some classes of machine learning models. Below, we justify why these rules may often be sufficient.

Let us start with Rule 1. Let $s$ be any univariate function. Then $s(x_i)$ is transparent because we can visualize it and create a mental model of its behavior. Let us now consider a transparent function $f$. As it is transparent, we should have a fairly good understanding of the properties of $f$. For instance, what range of values it attains, or whether it is monotonic for some subsets of the data. As we can

visualize $s$, it is reasonable to expect that we can infer these properties about $s \circ f$ as well. We can analyze $s$ and $f$ separately and then use that knowledge to analyze $s \circ f$.

Let us now justify Rule 2. Let $b \in \mathcal{B}$ be a binary operation and let $f$ and $g$ be transparent functions with non-overlapping sets of arguments. As these functions are transparent, we can understand their various properties. As they do not have any common variables, they act independently. Thus, we can combine them using the binary operation $b$ and directly use the previous analysis to understand the new model $b \circ (f, g)$. Thus, it is considered transparent. See Appendix D for a practical example.

Although Rule 2 seems like a strong constraint, many common closed-form equations used to describe natural phenomena can be put in a form that satisfies this rule. In particular, 82 out of 100 equations from Feynman Symbolic Regression Database [44] satisfy Rule 2. Thus, in many cases, the space of transparent models should be rich enough to find a good fit.

**Restricting the search space.** The current definition of SHAREs contains certain redundancies. For instance, it allows for a direct composition of two shape functions. This unnecessarily complicates the model as the composition of two shape functions is just another shape function (given that the class of shape functions is expressive enough). As any binary operation applied to a function and a constant can be interpreted as applying a linear function, we can remove the constants without losing the expressivity of SHAREs (given that the shape functions can model linear functions).

We can now use these two rules and the above observations to define transparent SHAREs.

**Definition 1.** A transparent SHARE is a SHARE that satisfies the following criteria:

- Any binary operator is applied to two functions with disjoint sets of variables.
- The argument of a shape function cannot be an output of another shape function, i.e., $s_1(s_2(x))$ is not allowed.
- It does not contain any numeric constants.

*Remark* 2. By this definition, any GAM is a transparent SHARE. This is consistent with the fact that GAMs are generally considered transparent models [17, 9].

Transparent SHAREs have some useful properties. For instance, there is no need to arbitrarily limit the size of the expression tree (as might be the case for many SR algorithms). The following proposition demonstrates some useful properties of SHAREs, including that the depth and the number of nodes of a transparent SHARE are naturally constrained.

**Proposition 1.** *Let $f : \mathbb{R}^n \to \mathbb{R}$ be a transparent SHARE. Then*

- *Each variable node appears at most once in the expression tree.*
- *The number of binary operators is $d - 1$, where $d$ is the number of variable nodes (leaves)*
- *The depth of the expression tree of $f$ is at most $2n$.*
- *The number of nodes in the expression tree of $f$ is at most $4n - 2$.*

*Proof.* Appendix A. □

For comparison, the expression tree of a GAM has $3n - 1$ nodes. That demonstrates that transparent SHAREs are not only naturally constrained, but even the largest possible expressions are not significantly longer than the expression for a GAM, even though it can capture much more complicated interactions. The immediate corollary of this proposition is useful for the implementation.

**Corollary 1.** *SHARE $f$ satisfies Rule 2 iff each variable appears at most once in its expression tree.*

# 5 SHAREs in Action

In this section, we perform a series of experiments to show how SHAREs work in action. First, we justify our claim that SHAREs extend GAMs (Section 5.1) and SR (Section 5.2). Finally, we show an example that cannot be fitted by GAM or by SR. For details about the experiments, see Appendix C.

**Implementation.** We use nested optimization to implement SHAREs. The outer loop employs a modified genetic programming algorithm (based on gplearn [41] used for symbolic regression), while the inner loop optimizes shape functions as neural networks via gradient descent. Although not a key contribution due to its limited scalability, this implementation demonstrates SHAREs' potential to outperform existing transparent methods and enhance interpretability, given more efficient optimization algorithms. We note that optimization of transparent models is usually harder than that of black boxes [35]. For further implementation details, see Appendix B.

## 5.1 SHAREs extend GAMs

As we discussed earlier, GAMs (without interactions) are examples of SHAREs. That means that, in a particular, if we have a dataset that can be modeled well by a GAM, SHAREs should also model it well. To verify this, we generate a semi-synthetic dataset inspired by the application of GAMs to survival analysis described in [18]. In this work, GAMs are used to model the risk scores of patients taking part in a clinical trial for the treatment of node-positive breast cancer. We choose three of the covariates considered and assume that the risk score (log of hazard ratio) can be modeled as a GAM of age, body mass index (BMI), and the number of nodes examined. We recreate the shape functions to resemble the ones reproduced in the original paper. Then we choose the covariates uniformly from the prescribed ranges and calculate the risk scores.

We fit SHAREs to this dataset and show the results in Figure 3. Each row shows the best equation with the corresponding number of shape functions and the shape functions of the equation with three shape functions are shown on the right side of the figure.

| #s | Equation | $R^2$ score |
|---|---|---|
| 0 | $y = \frac{x_{nodes}}{x_{age}}$ | -1.241 |
| 1 | $y = (x_{nodes} + x_{age})s_3(x_{bmi})$ | 0.623 |
| 2 | $y = s_2(x_{age}) + s_3(x_{bmi})$ | 0.855 |
| 3 | $y = s_1(x_{nodes}) + s_2(x_{age}) + s_3(x_{bmi})$ | 0.992 |
| 4 | $y = s_0\left(\frac{s_2(x_{age})s_3(x_{bmi})}{s_1(x_{nodes})}\right)$ | 0.992 |

Figure 3: Results of fitting SHAREs to the risk score data. Each row in the table shows the best found equation with the corresponding number of shape functions (#s). On the right side, shape functions from the fourth row compared to the ground truth.

We see that the equation in the fourth row achieves a high $R^2$ score. It is also in the desired form. When we plot the shape functions in Figure 3 we see that they match the ground truth well (the vertical translation is caused by the fact that shape functions can always be translated vertically).

## 5.2 SHAREs extend SR

**Torque equation.** Consider equation I.18.12 (Table 1) used to calculate torque, given by $\tau = rF\sin(\theta)$. We sample 100 rows from the Feynman dataset corresponding to this expression and we run our algorithm. Each row of the table in Figure 4 shows the best equation with the corresponding number of shape functions. The right side of the figure shows the shape functions of the equations in the second and fourth rows.

| # s | Equation | $R^2$ score |
|---|---|---|
| 0 | $\tau = \frac{F}{\theta}$ | -0.011 |
| 1 | $\tau = rFs_3(\theta)$ | 0.999 |
| 2 | $\tau = s_1(r)Fs_3(\theta)$ | 0.999 |
| 3 | $\tau = s_1(r)s_2(F)s_3(\theta)$ | 0.999 |
| 4 | $\tau = s_0(s_1(r)s_1(F)s_3(\theta))$ | 0.999 |

Figure 4: Equations found by fitting SHAREs to a torque equation $\tau = rF\sin(\theta)$. Each row in the table shows the best found equation with the corresponding number of shape functions (#s). Central panel: shape function from the second row compared to ground truth. Right panel: shape functions from the fourth row.

The equation that is symbolically equivalent to the ground truth is in the second row, $\tau = rFs_3(\theta)$. It achieves a nearly perfect $R^2$ score. By plotting $s_3$, we can verify that it matches $\sin$ function well (Figure 4, central panel).

**What are the shape functions of the longer equations?** Consider the expression in row 4 from the table in Figure 4, $\tau = s_1(r)s_2(F)s_3(\theta)$. It might look complicated because it contains three shape functions. But, if we inspect $s_1$ and $s_2$ (Figure 4, right panel), we see that they are linear functions. We can extract the line equations and put them into the found SHARE to get a simple expression $(-0.2r - 0.56)(0.34F + 0.78)s_3(\theta)$.

### 5.3 SHAREs go beyond SR and GAMs

We consider the following problem. Given $m$ grams of water (in a liquid or solid form) of temperature $t_0$ (in $°C$), what would be the temperature of this water (in a solid, liquid, or gaseous form) after heating it with energy $E$ (in calories). We restrict the initial temperature to be from -100 $°C$ to 0 $°C$. This is a relatively simple problem with only 3 variables but we will show that both GAMs and SR are not sufficient to properly (and compactly) model this relationship.

**GAMs.** First, we fit GAMs without interactions using EBM [25, 30]. The shape functions of EBM are presented in Figure 5 (right panel). The $R^2$ score on the validation set is 0.758. We can also see that the two of the shape function are very jagged. They do not seem to fit our definition of a simple univariate function. This makes it difficult to gain insight into the studied phenomenon.

**GA$^2$Ms.** Now, we fit GAMs with pairwise interactions [26], once again using the EBM algorithm. The shape functions of EBM are presented in Figure 8 in Appendix C.3. Although the $R^2$ score has been improved to 0.875, EBM's transparency is reduced even further by adding pairwise interactions.

**Symbolic Regression.** We fit symbolic regression using the PySR library [13]. We limit the complexity of the program to 40 and we present the results in the table in Figure 5. Only the most complex equations give us a performance comparable to a GAM with interactions: 0.867. The last equation from the table is shown below. We argue that its complexity hinders its transparency.

$$y = \frac{x_2}{\log(x_0)} - 1.72 e^{e^{\cos\left(\frac{0.0103 x_0}{x_1}\right)}} + 80.1 + 56.3 \cos\left(\log\left(\frac{0.58 x_0}{x_1} + 31.7 \cos\left(\frac{0.021 x_0}{x_1} + 0.93\right)\right)\right)$$

| Equation | Complexity | $R^2$ score |
|---|---|---|
| $y = 13.5 \log(E)$ | 4 | 0.384 |
| $y = \frac{0.193E}{m}$ | 5 | 0.485 |
| $y = 39.4 \log\left(\frac{E}{m}\right) - 141$ | 8 | 0.733 |
| Appendix C.3 | 17 | 0.768 |
| Appendix C.3 | 23 | 0.817 |
| Appendix C.3 | 33 | 0.840 |
| Appendix C.3 | 40 | 0.867 |

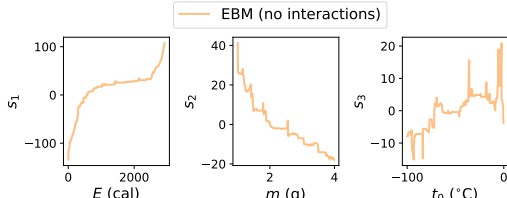

Figure 5: Left panel: Equations found by SR when fitted to the temperature data. The last four equations do not fit in the table; they are reproduced in Appendix C.3. Right panel: shape functions from the GAM fitted to the temperature dataset.

**SHAREs.** We finally fit SHAREs to the temperature data. The found expressions are shown in Figure 6. We immediately see a very good performance from all models apart from the one not using any shape functions at all. The scores are also much better than the scores achieved by GAMs (with or without interactions) and SR. Let us investigate the equation in the third row; the shape functions are presented in Figure 6 (right panel).

We note that the expression $s_1\left(\left(\frac{E}{m} + s_2(t_0)\right)\right)$ has a better performance than GAMs and SR, a more compact symbolic representation than SR, and simpler shape functions than GAM. This exemplifies how, by combining the advantages of GAMs and SR, we can address their underlying limitations. Let us see how this particular SHARE can aid in understanding the phenomenon it fits. We first recognize that $s_1$ is contingent on the energy-to-mass ratio, which is offset by a function of the initial temperature, $t_0$. As shown in Figure 6's right panel, $s_2$ appears linear, with an irregularity around $-40 °C$, which regularization may have eliminated. Replacing $s_2$ with an equivalent linear function and adjusting the equation gives us: $t = s_1\left(\frac{E}{m} + 0.507 t_0 + 24.973\right)$

Analyzing $s_1$, we find that without energy input, $\frac{E}{m} + 0.507 t_0 + 24.973$ ranges from $-26$ ($t_0 = -100$) to $25$ ($t_0 = 0$), aligning with the first linear part of the $s_1$ curve. Increasing energy per mass initially raises the temperature linearly to $0 °C$, then plateaus, characteristic of an ice-water mixture. When all ice melts, the temperature rises linearly again to $100 °C$, remaining constant until all water evaporates, after which steam temperature again increases linearly.

The shape functions also provide quantitative insights. The slopes of $s_1$'s linear parts approximate the specific heat capacities of ice, water, and steam. The constant parts' widths estimate the heat of fusion and vaporization. We compare these estimates from $s_1$ with the physical ground truth in Table 6 in Appendix C.3 (the same values were used for data generation).

| #s | Equation | $R^2$ score |
|----|----------|-------------|
| 0 | $t = m$ | -3.513 |
| 1 | $t = s_1\left(\frac{E+t_0}{m}\right)$ | 0.979 |
| 2 | $t = s_1\left(\left(\frac{E}{m} + s_2(t_0)\right)\right)$ | 0.999 |
| 3 | $t = s_1\left(\left(s_0\left(\frac{E}{m}\right)s_2(t_0)\right)\right)$ | 0.999 |
| 4 | $t = s_1\left(\left(s_3\left(\frac{s_0(E)}{m}\right) + s_2(t_0)\right)\right)$ | 0.999 |

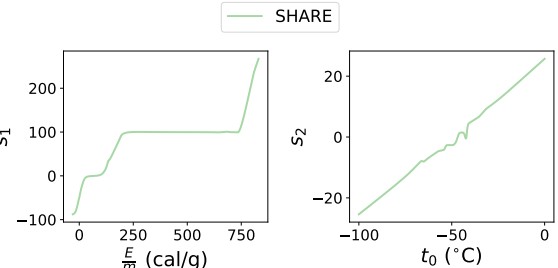

Figure 6: Equations found by fitting SHAREs to the temperature dataset. Each row in the table shows the best-found equation with the corresponding number of shape functions (#s). Right panel: shape functions of the found SHARE $s_1\left(\left(\frac{E}{m} + s_2(t_0)\right)\right)$ (third row).

## 6 Discussion

**Applications.** SHAREs can be beneficial in settings where transparent models are needed or preferred, such as risk prediction in healthcare and finance. They can also be useful in AI applications for scientific discovery (AI4Science). Currently, a lot of work in AI4Science is focused on developing better symbolic regression methods. We believe that for AI4Science to advance beyond the synthetic experiments based on simple physical equations, we need to add more flexibility to our models. Transparent SHAREs add this flexibility without compromising the comprehensibility.

**Limitations.** The current implementation of SHAREs is time-intensive and thus does not scale to bigger datasets. We are confident that further optimizations will enable wider adoption of this novel approach. We hope that future work will address the limitation of our implementation and will enhance the ability to fit SHAREs to even larger and more complex datasets.

## Acknowledgments

Krzysztof Kacprzyk is supported by Roche. This work was supported by Azure sponsorship credits granted by Microsoft's AI for Good Research Lab. We would like to thank Jeroen Berrevoets, Alan Jeffares, and Andrew Rashbass for their comments and feedback on an earlier manuscript.

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

## Table of supplementary materials

## A  Theoretical results

In this section, we provide proof of the properties listed in Proposition 1.

First, let us define *active variables* and a *subtree of a node*.

**Definition 2** (Active variables). Consider a SHARE $f : \mathbb{R}^n \to \mathbb{R}$ and its expression tree. The *set of active variables* of the tree (or of $f$) is the set of variables present in the tree.

For instance, a function $f(x_1, x_2, x_3) = x_1 + x_2$, represented as a tree with 3 nodes: +, $x_1$, $x_2$, has the set of active variables $\{x_1, x_2\}$ and a set of not active variables $\{x_3\}$.

**Definition 3** (Subtree of a node). Consider an expression tree $T$. For each node $A$ in $T$, we define the subtree of $A$ as a subtree of $T$ containing $A$ and all its descendants. We call $f_A$ the function represented by the subtree of $A$.

For the rest of the section, we assume that $f : \mathbb{R}^n \to \mathbb{R}$ is a transparent SHARE (according to Definition 1) and its expression tree is called $T$

### A.1  Variable nodes

Claim: Each variable node appears at most once in the expression tree. The maximum number of leaves is $n$.

*Proof.* Assume for contradiction there are two nodes, $A$ and $B$, describing the same variable $x$. Consider the lowest common ancestor of $A$ and $B$ called $C$. If $C$ was a shape function then the child of $C$ would be a lower common ancestor of $A$ and $B$. Thus $C$ is a binary function with children $C_1$ and $C_2$. Without loss of generality, assume that $A$ is in the subtree of $C_1$. Then $B$ has to be in the subtree of $C_2$ (otherwise $B$ would have to be in a subtree of $C_1$ and $C_1$ would be a lower common ancestor of $A$ and $B$). Thus the functions $f_{C_1}, f_{C_2}$ have a non-empty set of active variables (contains at least $x$). Thus $C$ is a binary operator applied to two functions with an overlapping set of active variables. Thus $f$ does not satisfy Rule 2 which contradicts $f$ being transparent.

Thus each variable node appears only once in the expression tree. As there are $n$ variables, there are at most $n$ variable nodes. This is the same as the number of leaves as variable nodes are the only kinds of leaves. $\qquad\square$

### A.2  Useful lemma

In the next proofs, the following lemma will be helpful.

**Lemma 1.** *Consider a node $A$ that corresponds to a binary operator. Let us call the children of $A$, $A_1$, and $A_2$. If $f_{A_1}$ has $a$ active variables $f_{A_2}$ has $b$ active variables then $f_A$ has $a + b$ active variables. Also $a, b < a + b$.*

*Proof.* The set of active variables of $f_A$ is the union of active variables of $f_{A_1}$ and $f_{A_2}$. As these functions have disjoint sets of active variables (because $f$ is transparent), the number of active variables of $f_A$ is just a sum of the numbers of active variables of $f_{A_1}$ and $f_{A_2}$. $\qquad\square$

### A.3  Number of binary operators

Claim: The number of binary operators is $d - 1$ where $d$ is the number of active variables of $f$.

*Proof.* Let us prove the following, more general, statement: Consider the node $B$. If the number of active variables of $f_B$ is $k$ then the number of binary operators in the subtree of $B$ is $k-1$.

We prove it by strong induction on $k$.

When $k = 1$ (we cannot have $k = 0$ as we do not have any constants) then the subtree of $B$ is either a variable node or a shape function with a node variable as a child. In both cases, the number of binary operators is $0$.

Let us assume the statement is true for all $m < k+1$. Assume that the subtree of $B$ has $k+1$ active variables. $B$ is either a shape function or a binary operator. If $B$ is a binary operator then it has two children. Let us call them $B_1$ and $B_2$. Let us denote the number of active variables of $f_{B_1}$ as $a$ and of $f_{B_2}$ as $b$. From Lemma 1, $k+1 = a+b$ and $a < k+1$, and $b < k+1$. From the induction hypothesis, the subtree of $B_1$ has $a-1$ binary operators and the subtree of $B_2$ has $b-1$ binary operators. Thus the subtree of $B$ has $(a-1)+(b-1)+1 = a+b-1 = k$ binary operators. If $B$ is a shape function then its child is a binary operator and the same argument follows.

By induction, if the number of active variables of $f_B$ is $k$ then the number of binary operators in the subtree of $B$ is $k-1$.

Now $f$ has $d$ active variables, so it has $d-1$ binary operators. $\square$

## A.4 Depth of the expression tree

Let $d$ be the number of active variables of $f$. By the previous result it has $d-1$ binary operators. That means that on the path from the root to the variable node there are at most $d-1$ binary operators. On this path, every pair of consecutive binary operators can be separated by at most one shape function (the same is true for a binary operator and a variable node). Thus the maximum number of nodes on the path from root to the variable node is a sum of $d-1$ (number of binary operators), $d-2$ (number of shape functions between the operators), $1$ (shape function between the operator and the variable node), $1$ (shape function as a root), $1$ (the variable node itself). This gives a total of $(d-1)+(d-2)+1+1+1 = 2d$. Thus the maximum depth of the tree is $2d$. As $d \leq n$, we get that the maximum depth of the tree is $2n$.

## A.5 Size of the expression tree

Claim: The number of nodes in a tree is at most $4n-2$.

*Proof.* Let us prove the following, more general statement: Consider a node $A$. If the number of active variables of $f_A$ is $k$ then the maximum number of nodes in the subtree of $A$ is $4k-2$ if $A$ is a shape function and $4k-3$ otherwise.

Let us prove it by strong induction on $k$.

Consider $k = 1$. The subtree of $A$ is either a variable node, or a shape function with a variable node as a child. The number of nodes is either $1$ if it is a variable node or $2$ if it is a shape function. As $4 \times 1 - 2 = 2$ and $4 \times 1 - 3 = 1$, the base case is satisfied.

Let us assume the statement is true for all $m < k+1$.

Consider a node $A$ whose subtree has $k+1$ active variables. If $A$ is a binary operator then it has two children $A_1$ and $A_2$. Their subtrees have respectively $a$ and $b$ active variables. From Lemma 1, $a+b = k+1$. By the induction hypothesis, the maximum number of nodes in the subtree of $A_1$ is $4a-2$ and $4b-2$ in the subtree of $A_2$. Thus the maximum number of nodes in the subtree of $A$ is $(4a-2)+(4b-2)+1 = 4(a+b)-3 = 4(k+1)-3$. This proves one part of the claim. If $A$ is a shape function then its child is a binary operator with $k+1$ active variables. But we have just proved that the subtree of this operator has at most $4(k+1)-3$ nodes. That means that the maximum number of nodes in the subtree of $A$ is $4(k+1)-3+1 = 4(k+1)-2$ as required.

The claim is true by induction. Now we want to show that such a tree always exists. Consider a binary operator node $A_1$ whose subtree has $k$ active variables $\{x_1, \ldots, x_k\}$. Let its children be two shape functions $B_1$ and $C_1$. Let the child of $C_1$ be a variable node corresponding to $x_1$. Let the child of $B_1$ be a binary operator $A_2$. We repeat the process. In general, binary operator node $A_i$ has two children $B_i$ and $C_i$. The child of $C_i$ is a variable node corresponding to $x_i$ and the child of $B_i$ is the binary

operator $A_{i+1}$. We can repeat the process until $i = k - 1$. At this point the child of $B_{k-1}$ needs to be a variable node corresponding to $x_k$. Overall, we have $k - 1$ binary operators $A_1, \ldots, A_{k-1}$. $k - 1$ shape functions $B_1, \ldots, B_{k-1}$, $k - 1$ shape functions $C_1, \ldots, C_{k-1}$, and $k$ variable nodes. Thus the total number of nodes is $3(k-1) + k = 4k - 3$. If the first node is a shape function then its child is the binary operator node $A_1$ and the total number of nodes is $4k - 2$.

As the number of active variables in the whole tree is less than $n$, then the maximum number of nodes is $4n - 2$. $\qquad\square$

# B    Implementation

We implement SHAREs using nested optimization. The outer loop is a modified genetic programming algorithm (based on gplearn [41] that is used for symbolic regression) that finds a symbolic expression with placeholders for the shape functions and the inner loop optimizes the shape functions. We implemented the shape functions as neural networks and optimize the model using a gradient descent algorithm.

## B.1    Modifications to the genetic algorithm

Gplearn is a symbolic regression algorithm that represents equations as expression trees and uses genetic programming to alter the equations (programs) from one generation to the next one based on their fitness score. We modify this algorithm so that the found expressions contain placeholders for the shape function. To compute the fitness of an expression the whole equation is fitted to the data and the shape functions are optimized using gradient descent. To guarantee that all equations are transparent (i.e., they satisfy Definition 1) we change the way the initial population is created and modify some of the rules by which the equations evolve. We describe the details of the modifications below.

**Initial population.** We disable the use of constants and allow only binary operations in $\mathcal{B}$ and shape functions. We grow the expression trees at random starting from the root. The next node is chosen randomly and constrained such that: a) if the parent node is a shape function then the child cannot be a shape function, and b) no variable can appear twice in the tree. By the Corollary 1, the second condition is equivalent to satisfying Rule 2.

**Crossover.** We call the variables present in a tree *active variables*. During crossover, we select a random subtree from the program to be replaced. We take the union between the active variables in the subtree and the variables that are not active in the whole program. A donor has a subtree selected at random such that its set of active variables is contained in the previous set. This subtree is inserted into the original parent to form an offspring. This guarantees that no variable appears twice in the offspring.

**Subtree mutation.** We perform the same procedure as in crossover but instead of taking a subtree from a donor, we create a new program with variables from the allowed set.

**Point mutation.** This procedure selects a node and replaces it for a different one. A binary operation is replaced by a different binary operation. Shape functions are not replaced. All variables that are supposed to be replaced are collected in a set. This set is enlarged by the variables that are not active. For each mutated node the variable is drawn from this set without replacement.

Reproduction and hoist mutation has not been altered. For more details about the genetic programming part of the algorithm, please see the official documentation of gplearn.

**Binary operations $\mathcal{B}$.** We choose the set of binary operations to be $\mathcal{B} = \{+, \times, \div\}$ (we remove "$-$" to remove redundancy and reduce the search space.

## B.2    Optimization of the shape functions

**Shape functions $\mathcal{S}$.** We choose the set of shape functions $\mathcal{S}$ to be a family of Neural Networks with 5 hidden layers, each with a width of 10. Each layer, excluding the last one, is followed by an ELU activation function. We apply batch normalization before the input layer.

**Dataset vs. batch normalization.** We do not perform any kind of normalization on the whole dataset before training. This is driven by the fact that we want to use the form of the equation for analysis, debugging or gaining insights. Dataset normalization makes the feature less interpretable by, de facto, changing the units in which they are measured. Moreover, such normalization might make certain invariances more difficult to detect. Translational or scale invariances are present in many physical systems and, in fact, have been used to discover closed-form expressions from data [44]. Consider equation $(x_2 - x_1)^2$. The value of the expression does not depend directly on the values of $x_1$ and $x_2$ but rather on their difference $x_2 - x_1$. Detecting this relationship is important for both creating equations with interpretable terms and for pruning the search space. As we tend to use a consistent and familiar set of units, we want to capitalize on that as much as we can. However, features on different scales make neural networks (and other machine learning algorithms) notoriously difficult to train. That is why we perform batch normalization before passing the data to a shape function. That allows to perform a series of binary operations in the original units before a shape function is applied. This is what happens in the example in Section 5, where given the energy and the mass of the substance, their ratio (energy per 1 gram) is discovered to be a more meaningful feature.

### B.3 Pseudocode and a diagram

**Block diagram.** The training procedure for SHAREs implemented with symbolic regression and neural networks is depicted in Figure 7.

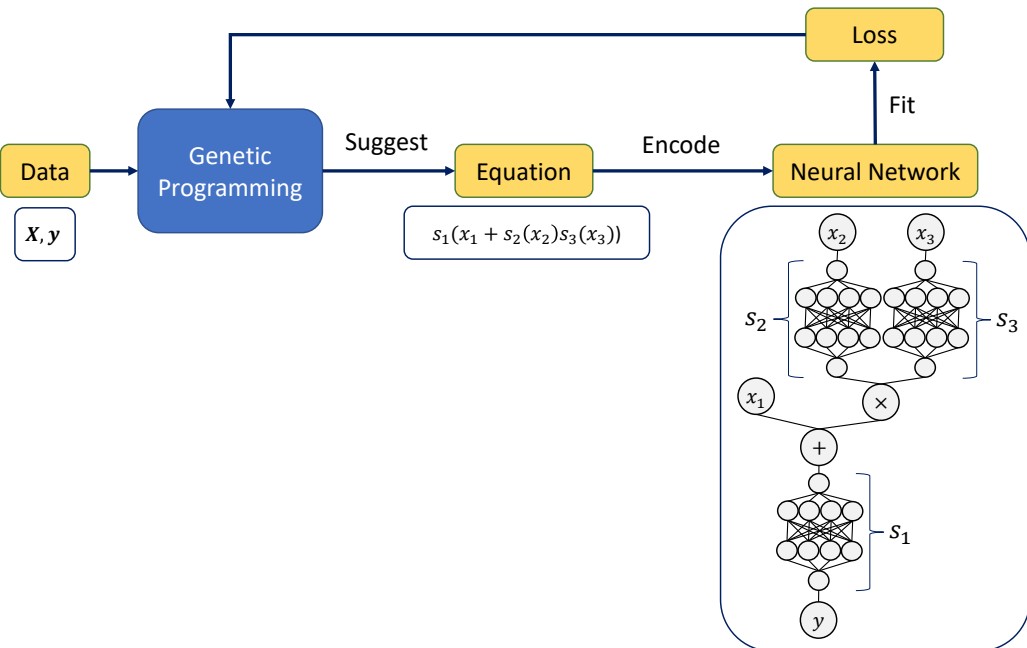

Figure 7: This figure shows a block diagram depicting our implementation of SHAREs

**Pseudocode.** The pseudocode for SHAREs implemented with symbolic regression and neural networks is described in Algorithm 1.

## C Experiments

### C.1 Hyperparameters

**gplearn.** Gplearn hyperparameters used for experiments are presented in Table 2.

**Optimization of shape functions.**

Hyperparameters used for optimizing the shape function are presented in Table 3.

**Algorithm 1** SHARE implemented using symbolic regression and neural networks

---

**Input:** Data $\boldsymbol{X}, \boldsymbol{y}$
**Input:** Symbolic regression optimization algorithm $\mathcal{O}_{\text{symbolic}}$
**Input:** Gradient-based optimization algorithm $\mathcal{O}_{\text{gradient}}$
**Output:** SHARE
  **procedure** LOSS($f_e$)
    Encode expression $f_e$ as a neural network $f$
    $f \leftarrow \mathcal{O}_{\text{gradient}}\left(||\boldsymbol{y} - f(\boldsymbol{X})||_2^2\right)$
    **return** $||\boldsymbol{y} - f(\boldsymbol{X})||_2^2$
  **end procedure**
  $f_e = \mathcal{O}_{\text{symbolic}}(\text{LOSS})$
  **return** $f_e$

---

Table 2: Gplearn hyperparameters used in the experiments

| Hyperparameter | Value |
|---|---|
| Population size | 500 |
| Generations | 10 |
| Tournament size | 10 |
| Function set | $+, \times, \div,$ *shape* |
| Constant range | None |
| p_crossover | 0.4 |
| p_subtree_mutation | 0.2 |
| p_point_mutation | 0.2 |
| p_hoist_mutation | 0.05 |
| p_point_replace | 0.2 |
| Parsimony coefficient | 0.0 |

Table 3: Hyperparameters used in shape function optimization

| Hyperparameter | Value |
|---|---|
| Algorithm | Adam [21] |
| Maximum num. of epochs | 1000 |
| Early stopping patience | 10 (evaluated every 10 epochs) |
| Early stopping tolerance | 0.001 |
| Learning rate | Tuned automatically for each equation |

**PySR.** PySR hyperparameters used in the experiments are presented in Table 4.

Table 4: PySR hyperparameters

| Hyperparameter | Value |
|---|---|
| Binary operations | $+, -, \times, \div$ |
| Unary operators | $\log, \exp, \cos$ |
| maxsize | 40 |
| populations | 30 |
| niterations | 400 |
| population_size | 50 |

**EBM.** EBM hyperparameters used in the experiments are presented in Table 5

Table 5: EBM hyperparameters

| Hyperparameter | Value |
|---|---|
| max_bins | 256 |
| max_interaction_bins | 32 |
| binning | quantile |
| interactions | 0 or 3 |
| outer_bags | 8 |
| inner_bags | 0 |
| learning_rate | 0.01 |
| validation_size | 0.15 |
| early_stopping_rounds | 50 |
| early_stopping_tolerance | 0.0001 |
| max_rounds | 5000 |
| min_samples_leaf | 2 |
| max_leaves | 3 |

## C.2 Data generation

**Risk scores dataset.** [18] describes a process of applying GAMs to a dataset of patients taking part in a clinical trial for the treatment of node-positive breast cancer. In the paper, three shape functions that resulted from this fitting are presented (Figure 1). To generate the risk scores data we use for experiments in Section 5, we use these plots to create similar-looking functions using BSplines. We sample uniformly each of the covariates from their corresponding ranges, i.e., $x_{nodes} \in (0, 50)$, $x_{age} \in (45, 70)$, and $x_{bmi} \in (17, 45)$. We then apply the created shape functions to these covariates and add their values together. We create 200 samples. Half of them is used for training and the other half for validation.

**Temperature dataset.** Data used in temperature experiments in Section 5 was generated by simulating the temperature of water based on the laws of physics and constants shown in Table 6. $m$ was uniformly sampled from $(1, 4)$ and $t_0$ was sampled uniformly from $(-100, 0)$. The energy $E$ was calculated by first sampling energy per mass uniformly from $(1, 800)$ and then multiplying it by the mass $m$. If the energy was uniformly sampled directly then the ratio $\frac{E}{m}$ would have very non-uniform distribution which would inhibit learning. We draw 2000 samples. Half of them is used for training and the other half for validation.

## C.3 Additional results

**GA$^2$M fitted to the temperature dataset.** We present the shape functions of GA$^2$M fitted to the temperature dataset in Section 5.

**Equations from PySR fitted to the temperature dataset.** We present the equations found by PySR when fitted to the temperature dataset in Section 5. These equations did not fit into the table with the results.

$$y = 74.0 \cos\left(\log\left(\frac{0.739E}{m} + 19.1\right)\right) + 39.1 + \frac{t_0}{E} \tag{4}$$

$$y = \frac{t_0}{\log(E)} + 65.0 \cos\left(\log\left(\frac{E}{m} + 41.5 \cos\left(\frac{0.0275E}{m}\right)\right)\right) + 50.9 \tag{5}$$

$$y = -1.63 e^{e^{\cos\left(\frac{0.0101E}{m}\right)}} + 58.6 \cos\left(\log\left(\frac{0.653E}{m} + 27.1 \cos\left(\frac{0.0261E}{m}\right)\right)\right) + 67.6 \tag{6}$$

$$y = \frac{x_2}{\log(x_0)} - 1.72 e^{e^{\cos\left(\frac{0.0103E}{m}\right)}} + 56.3 \cos\left(\log\left(\frac{0.582E}{m} + 31.7 \cos\left(\frac{0.0214E}{m} + 0.925\right)\right)\right) + 80.1 \tag{7}$$

**Water properties** extracted from found shape functions in Section 5 can be found in Table 6.

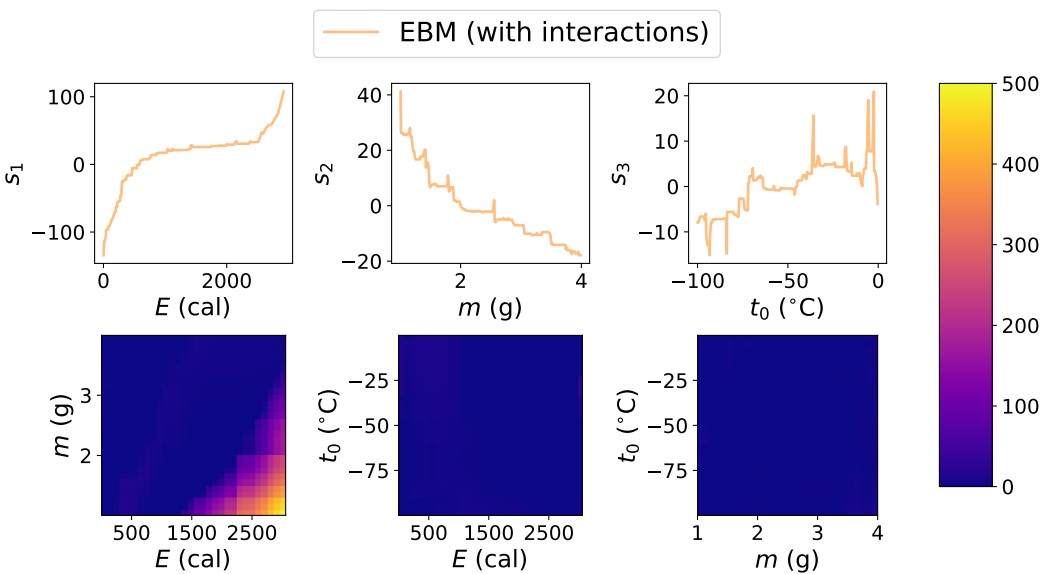

Figure 8: Shape functions from the GA$^2$M fitted to the temperature dataset

Table 6: Properties of water extracted from shape function $s_1$ compared to the ground truth.

| Property | From $s_1$ | Ground truth |
|---|---|---|
| Spec. heat cap. of ice ($\frac{cal}{g°C}$) | 0.46 | 0.50 |
| Spec. heat cap. of water ($\frac{cal}{g°C}$) | 0.98 | 1.00 |
| Spec. heat cap. of steam ($\frac{cal}{g°C}$) | 0.48 | 0.48 |
| Heat of fusion ($\frac{cal}{g}$) | 82.81 | 79.72 |
| Heat of vaporization ($\frac{cal}{g}$) | 589.88 | 540.00 |

## C.4 Additional experiments

We performed an additional experiment on the Boston House Prices dataset [16], and Concrete Compressive Strength dataset [50] from the UCI repository. The Boston dataset contains 13 features, including both numerical and categorical variables (note that standard Symbolic Regression is not well suited to categorical variables). We compare SHAREs with Linear Regression, GAMs, and XGBoost. We consider three kinds of GAMs: GAMs with splines (GAM-S) as implemented in the PyGAM library [38], EBMs without interactions (EBM-1) [25], and EBMs with pairwise interactions (EBM-2) [26]. Both are implemented in InterpretML package [30]. The results can be seen in Table 7. We present the shape functions found by SHAREs in Figure 9, by GAM-S in Figure 10, by EBM-1 in Figure 11, and by EBM-2 in Figure 12. Our analysis reveals that SHARE surpasses all other white box methodologies in performance on the Boston dataset, while maintaining superior interpretability of its shape functions. When applied to the Concrete dataset, SHARE is superior to Linear Regression and GAM-S, both of which are transparent algorithms. It is noteworthy that EBM-1 and EBM-2 also have a much more complex shape functions (Figure 11 and Figure 12).

## C.5 Computation time

The experiments were performed on 12th Gen Intel(R) Core(TM) i7-12700H with 64 GB of RAM. The total time of running all experiments was around 1h 45m.

Table 7: Results on two UCI datasets. The higher the score the better.

| Algorithm | Boston ($R^2$ score) | Concrete ($R^2$ score) |
|---|---|---|
| Linear Regression | 0.711 (0.072) | 0.593 (0.068) |
| GAM-S | 0.817 (0.054) | 0.877 (0.019) |
| EBM-1 | 0.812 (0.064) | 0.901 (0.015) |
| EBM-2 | 0.838 (0.058) | 0.928 (0.013) |
| XGBoost | 0.879 (0.048) | 0.929 (0.018) |
| SHARE | 0.840 (0.073) | 0.890 (0.018) |

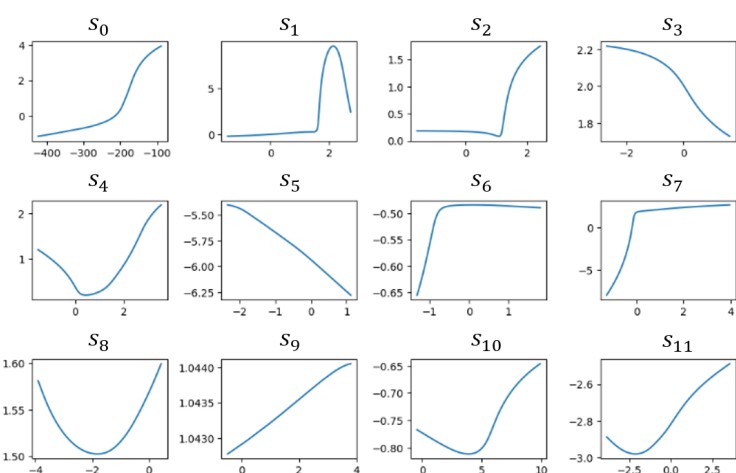

Figure 9: SHARE shape functions found when fitted to the Boston House Prices dataset

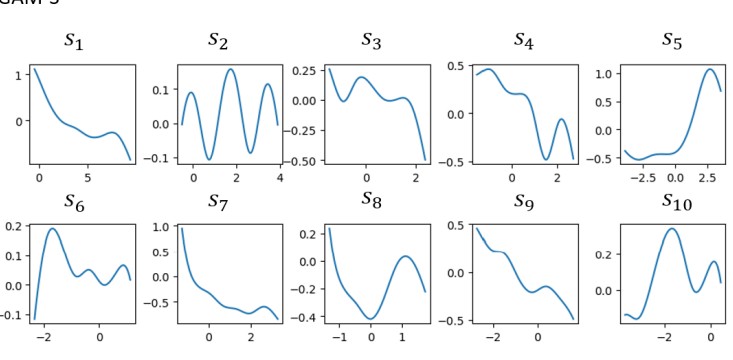

Figure 10: GAM-S shape functions found when fitted to the Boston House Prices dataset

## C.6 Software used

We use PySR [13] to run Symbolic Regression experiments.

We use the implementation of EBM [25, 26] available in the InterpretML package [30].

We use PMLB [31] package to access the Feynman Symbolic Regression Dataset.

## C.7 Licenses

The licenses of the software used in this work are presented in Table 8

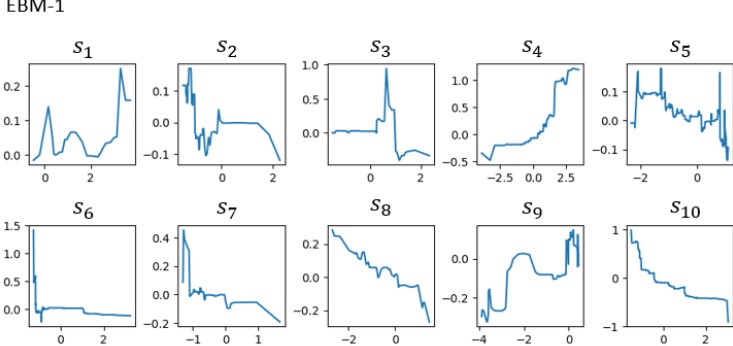

Figure 11: EBM-1 shape functions found when fitted to the Boston House Prices dataset

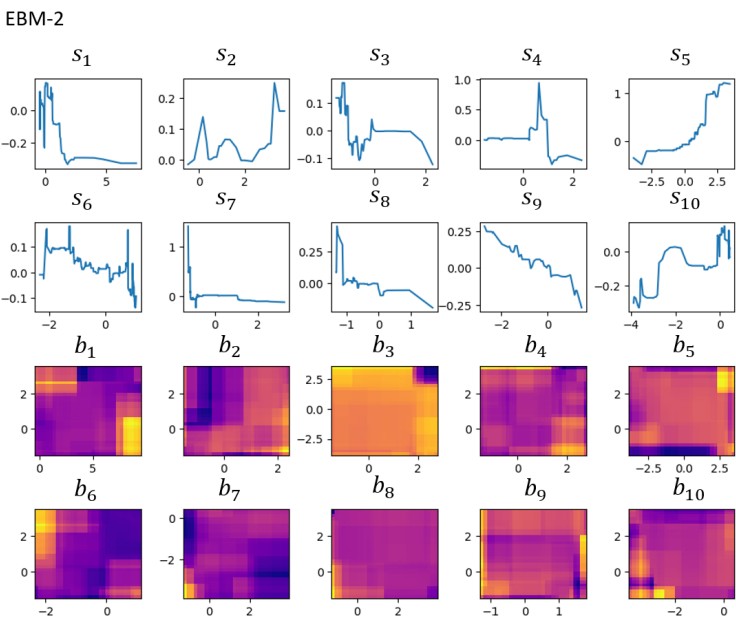

Figure 12: EBM-2 shape functions found when fitted to the Boston House Prices dataset

# D Discussion

## D.1 Equivalent solutions

As SHAREs are defined by their symbolic representation, it is possible that there are two different symbolic expressions that describe the same equation (especially as shape functions are flexible). We address this problem in three ways that tackle three types of equivalence relations:

1. In our implementation, we restrict the set of binary operations to $\{+, \times, \div\}$ (Table 2). As subtraction ("$-$") is not included, we do not get the equivalence $s_1(x_1) + s_2(x_2) = s_1(x_1) - s_2'(x_2)$ by $s_2' = -s_2$.
2. A common way two mathematical expressions can be equivalent is through the *distributive property*, i.e., $x_1 \times (x_2 + x_3) = x_1 \times x_2 + x_1 \times x_3$. However, thanks to our definition of transparency, the second expression will never appear in our search space (because the binary operators need to be disjoint).
3. We do not allow constants in transparent SHAREs to prevent equivalence of the type: $s(x) = s'(x \times a)$ for any $a \in \mathbb{R} \setminus \{0\}$ and $s'(x) = s(\frac{x}{a})$

Table 8: Software used and their licenses

| Software | License |
|---|---|
| gplearn | BSD 3-Clause "New" or "Revised" License |
| scikit-learn | BSD 3-Clause "New" or "Revised" License |
| numpy | liberal BSD license |
| pandas | BSD 3-Clause "New" or "Revised" License |
| scipy | liberal BSD license |
| python | Zero-Clause BSD license |
| PySR | Apache License 2.0 |
| interpret | MIT License |
| pmlb | MIT License |
| pytorch | BSD-3 |
| pytorch lightning | Apache License 2.0 |
| tensorboard | Apache License 2.0 |
| py-xgboost | Apache License 2.0 |
| pyGAM | Apache License 2.0 |

Another type of equivalence relation can arise from the use of exponential and logarithmic functions. For instance, $s_0(s_1(x_1) + s_2(x_2))$ can be represented as $s'_0(s'_1(x_1) \times s'_2(x_2))$ by taking $s'_1 = e^{s_1}, s'_2 = e^{s_2}, s'_0(x) = s_0(\log(x))$. This can be observed in rows 3 and 4 in the table in Figure 6. First, we note that, in these cases, the fact that these equivalent expressions (rather than something completely different) appear in the table is a sign of the robustness of our method. Even if we allow for more flexibility, we get expressions that are *meaningful* and can be *transformed* into simpler representations. Second, the other forms of expressions require more shape functions. Thus, as they have the same predictive power, the user can choose the one with a smaller number of shape functions for better understanding.

## D.2 Limitations of the current implementation

The current implementation of SHAREs is time-intensive and thus does not scale to bigger datasets - thus it is not a main contribution of our paper. We hope that future work will address the limitation of our implementation and will enhance the ability to fit SHAREs to even larger and more complex datasets.

The main bottleneck comes from the nested optimization and the necessity of fitting a separate neural network for every equation. Nevertheless, we want to highlight a few things we have done to make this problem more tractable:

1. The constants are *not* optimized by random mutations but implicitly by fitting the shape functions using gradient descent.
2. By considering only transparent SHAREs, we efficiently reduce to search space of expressions. By Proposition 1, the size of a SHARE is bounded by $4n - 2$, linear in the number of variables.
3. During training we cache the scores for found expressions so that they can be retrieved if they appear once again during the evolution.

## D.3 Taking advantage of units

Variables in the dataset are often expressed in certain units. These units often provide a lot of information and are frequently used in SR algorithms. Either explicitly [44] or implicitly by assuming that certain arithmetic operations make sense. For instance, adding two variables makes little sense if they are not measured in the same units. Of course, units may easily be changed by an affine transformation, but such transformations increase the length of the equation. As many SR algorithms penalize based on the length of the expression, *the change of units changes the score of the expression*. On the other hand, the datasets often combine observations of very different phenomena that might be measured in wildly different units.

So, we want to use the information about units when possible but we do not want to depend on it. This was one of the motivations behind SHAREs. They allow to use the binary operations on the

raw variables, capitalizing on the units in which they were described but also allow to first pass a variable through a shape function that can transform the variable. In certain cases (such as an affine transformation) this corresponds to a change of units.

## D.4   Related works

As SHAREs are very much related to GAMs and SR, we provide an overview of these two types of models below.

**Generalized Additive Models.** Generalized Additive Models (GAMs) were introduced in [17]. Initially, splines and other simple parametric functions were used as shape functions. In recent years many different classes of functions were proposed, including, tree-based, gradient boosted [25], deep neural networks [3, 33] or neural oblivious decision trees [10]. GAMs have also been extended to include pairwise interaction that can be visualized using heat maps [26]. A few neural network architectures closely tied to GAMs have also been proposed [42, 49].

**Symbolic Regression.** Symbolic Regression (SR) is a branch of machine learning that aims to construct a model in the form of a closed-form expression. Traditionally Genetic Programming [24] has been used for this task [7, 37, 14, 41]. Recently, this area attracted a lot of interest from the deep learning community. Neural networks have been used to prune the search space of possible expressions [44, 43] or to represent the equations directly by modifying their architecture and activation functions [27, 36]. A different approach is proposed in [6], where a neural network is pre-trained using a curated dataset. A similar approach is employed in [15, 20]. Methods using deep reinforcement learning [32] have also been proposed, as well as a hybrid of the two approaches [19].

**Complexity metrics used in symbolic regression.** Most of the metrics used in symbolic regression are based on the "size" of the equation. That includes the number of terms [41], the depth of a tree [13, 32], and the description length [43]. Pretrained methods often control the complexity of the generated equations by constraining the training set using the above methods [6]. Methods that directly represent the equation as a neural network (with modified activation functions) employ sparsity in the network weights [36]. Although these metrics are often correlated with the difficulty of understanding a particular equation, size does not always reflect the equation's complexity as it disregards its semantics. Some approaches try to address this issue. [46] introduces a metric based on "order of nonlinearity" with the assumption that nonlinearity measures the complexity of the function. Although simpler models tend to be more linear, and nonlinearity may be important for generalization properties, it is not clear how it aids in model understanding. Similarly, [45] uses curvature as an inspiration for their metric. Although curvature may be well-suited for characterizing bloat and overfitting, it does not directly relate to how the model is understood. [23] introduces a metric that is supposed to reflect the difficulty in understanding an equation. However, the exact rules chosen to calculate the complexity seem arbitrary. For instance, it is not clear why applying some well-known functions (such as $\sin$, $\log$) increases the complexity in a widely different manner than squaring or taking a square root. We also dispute the motivating example that demonstrates that $e^{\sin\sqrt{x}}$ is nearly 4000 times more complex than $7x^2 + 3x + 5$. It is not clear under what assumptions such a result would be intuitive.

## D.5   Meaning of the word transparent

In our paper, we have used a widely accepted term: *transparent* [5]. However, other terms could also be used. That includes: *inherently interpretable* [34], *intrinsically interpretable* [47], *intelligible* [25, 26], or *white boxes* [30]. For this paper, we assume all of these terms refer to the same class of models.

## D.6   Choice of equations

We choose equations in Section 2.2 so that they represent a variety of non-additive interactions between variables. Equations I.8.14 and I.29.16 describe the Euclidean distance in two dimensions and the Law of Cosines. Both of them involve a square root of a sum of terms. Equations I.12.2, I.18.12, I.32.5, II.2.42 can be used to describe an electric force between charged bodies, a torque, a rate of radiation of energy, and a heat flow. All of them are either products or fractions of products. Equations I.6.20b and I.40.1 describe a Gaussian distribution and a particle density. They both contain

exponential functions. Lastly, equation I.12.11 describes a Lorentz force via a sum of products, one of which contains a trigonometric function.

## D.7    Complexity of univariate functions.

Rule 1 in Section 4 is based on the assumption that any univariate function can be understood by plotting it. This assumption is tacitly made in many works on GAMs that introduce algorithms producing sometimes very complicated shape functions [25, 9, 3]. However, in some scenarios this assumption is too strong and some recent works introduce GAMs that take into account the complexity of shape functions [2]. Rule 1 can be modified to include these stronger assumptions. In our implementation, we use neural networks, which are known for their expressiveness, and we took a few design choices that made them simple in practice. We believe their simplicity is a result

- choosing a smooth ELU activation function instead of ReLU or ExU [3] that encourage more jagged functions
- employing early stopping when training the neural networks to prevent over-fitting
- optimizing using backprop - models learned that way were shown to be biased toward smooth solutions [8]

If these techniques are not sufficient to arrive at a desirable level of simplicity, a user can add a regularisation term to the loss function while training the neural networks. Our formulation of SHAREs allows for different kinds of shape functions, and thus, there is a way to enforce simplicity by using splines instead of neural networks. This assumes that our definition of simplicity concerns the smoothness and the number of inflection points. By choosing the number of knots, we can choose the level of simplicity we desire.

## D.8    Rules: practical example.

We propose a practical example when Rule 1 and Rule 2 are satisfied. Let us assume that our definition of transparency concerns (maybe, among other things) understanding the set of possible values the expression outputs given a particular set of inputs. We characterize the set of inputs by specifying an interval for every feature. That is, we are interested in the range of values of $f(x_1, \ldots, x_n)$ where $x_1 \in [a_1, b_1], \ldots x_n \in [a_n, b_n]$. Let us fix the input intervals. Let us assume that an expression $f(x_1, \ldots, x_n)$ is transparent if we can easily find an interval $[c, d]$ that is an image of this function for the specified inputs. Clearly, $x_i$ is transparent. But so is $f(x_i)$ for any univariate $f$ as long as we are able to characterize the extrema of $f$ at interval $[a_i, b_i]$. Let us assume that a given $f$ is transparent, i.e., we know its image is an interval $[c, d]$. Then, we can compute the image of $s \circ f$ (as long as we are able to characterize the extrema of $s$ at interval $[c, d]$). Thus, we can see how Rule 1 conforms to our notion of transparency. As Rule 2 requires the functions to be transparent and have disjoint sets of arguments, we can easily calculate the range of the whole model by applying the interval arithmetic. The example above demonstrates that our rules have practical application for a broad set of shape functions—the only requirements are being continuous and having easily identifiable extrema at given intervals (which can be found by plotting the function).

