# OpenReview forum: "Shape Arithmetic Expressions"
_NeurIPS.cc/2023/Workshop/AI4Science — NeurIPS2023-AI4Science Poster_

### Meta-Review · Area_Chair_h8oD · 2023-10-26

**Recommendation:** Accept (Poster)
**Confidence:** 3

**Metareview:**

This paper introduces Shape Arithmetic Expressions (SHAREs), which combine the flexibility of Generalized Additive Models (GAMs) with the ability to capture complex feature interactions found in mathematical expressions. The paper addresses the limitations of Symbolic Regression (SR) and GAMs, emphasizing that SR struggles with non-closed-form expressions, while GAMs cannot model complex interactions. To address these issues, SHAREs are proposed as a more flexible and interpretable modeling approach. The paper also defines rules for constructing transparent SHAREs. A comparison against SR and GAM is performed across various settings to demonstrate improved performance as a case study basis. However, their complexity and scalability should be further explored, and comparative analyses are needed for a comprehensive evaluation. The work should be of broad interest to the community. Recommendation: Poster